# Coupled Diffusion Sampling for Training-free Multi-view Image Editing

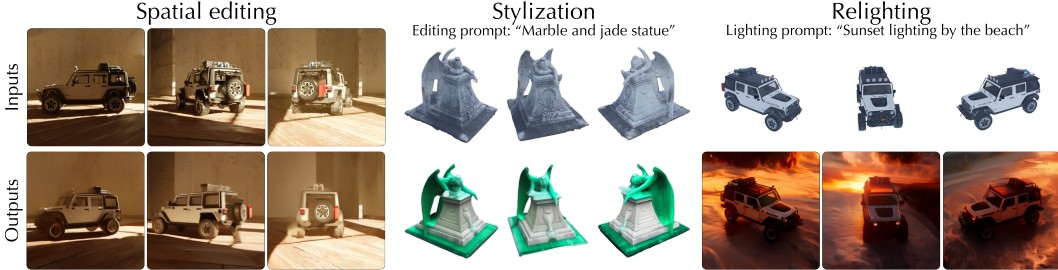

Figure 1: **Applications of coupled diffusion sampling.** Our approach enables lifting off-the-shelf 2D editing models into multi-view by combining the sampling process of 2D diffusion models with multi-view diffusion models to produce view-consistent edits. Here we showcase example view-consistent results using a 2D spatial editing model, stylization, and text-based relighting.

## ABSTRACT

We present an inference-time diffusion sampling method to perform multi-view consistent image editing using pre-trained 2D image editing models. These models can independently produce high-quality edits for each image in a set of multi-view images of a 3D scene or object, but they do not maintain consistency across views. Existing approaches typically address this by optimizing over *explicit* 3D representations, but they suffer from a lengthy optimization process and instability under sparse view settings. We propose an *implicit* 3D regularization approach by constraining the generated 2D image sequences to adhere to a pre-trained multi-view image distribution. This is achieved through *coupled diffusion sampling*, a simple diffusion sampling technique that concurrently samples two trajectories from both a multi-view image distribution and a 2D edited image distribution, using a coupling term to enforce the multi-view consistency among the generated images. We validate the effectiveness and generality of this framework on three distinct multi-view image editing tasks, demonstrating its applicability across various model architectures and highlighting its potential as a general solution for multi-view consistent editing.

## 1 INTRODUCTION

Diffusion-based image editing models have demonstrated unprecedented realism across diverse tasks via end-to-end training. These include object relighting (Jin et al., 2024; Magar et al., 2025; Zhang et al., 2025a), spatial structure editing (Wu et al., 2024b; Mu et al., 2024; Alzayer et al., 2025b; Vavilala et al., 2025), and stylization (Zhang et al., 2023). However, collecting and curating 3D data is significantly more costly than working with 2D data. As a result, recent research has explored test-time optimization methods for multi-view editing that leverage pre-trained 2D image diffusion models (Poole et al., 2023; Haque et al., 2023).

Lifting 2D image editing models directly to the 3D multi-view domain is non-trivial, primarily due to the difficulty in ensuring 3D consistency across different viewpoints. To address this, most existing methods (Haque et al., 2023; Jin et al., 2024) rely on explicit 3D representations, *i.e.,* NeRF (Mildenhall et al., 2020) or 3D Gaussian Splatting (Kerbl et al., 2023). Despite achieving promising results in certain scenarios, these methods typically require time-consuming optimization and dense input view coverage. This significantly limits their applicability to real-time, real-world scenarios.

Can we directly extend the capabilities of 2D image editing models to the multi-view domain *without* relying on explicit 3D representations or incurring additional training overhead? We answer this question affirmatively by introducing a novel diffusion sampling method — "coupled diffusion sampling". As shown in Fig. 1, our approach enables multi-view consistent image editing across diverse applications, including multi-view spatial editing, stylization, and relighting.

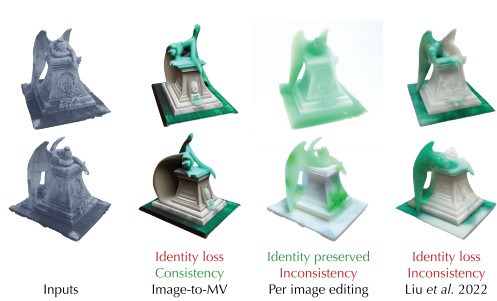

Figure 2: **Limitations of baselines.** Using a pre-trained image-to-multiview model conditioned on an edited image, can only be faithful to that single image but not the rest of the input views. On the other hand, editing each image individually with the 2D model produces highly inconsistent results. While prior work (Liu et al., 2022) proposes a method to compose diffusion models within the same domain, we find that their approach produces flickering results and cannot guarantee being faithful to the input views.

As shown in Fig. 2, sampling from two diffusion models independently yields samples that are inconsistent across views. Conditioning a multi-view model using a single edited image, however, fails to preserve identity and align with the editing objective across all views. While prior work (Liu et al., 2022; Du et al., 2023) explored combining diffusion models within a modality, we observe that such approaches do not maintain multi-view consistency and can stray from the editing objective. Our approach is motivated by the observation that, any sequence of images generated by a pre-trained multi-view image diffusion model inherently exhibit multi-view consistency. To this end, we embrace an implicit 3D regularization paradigm by leveraging scores estimated from multi-view diffusion models during the diffusion sampling process. Specifically, for any multi-view image editing task with a pre-trained 2D model, we couple it with a foundation multi-view diffusion model and perform sampling under dual guidance from both models. This process ensures that the resulting samples satisfy both the editing objective and multi-view 3D consistency, yet without any additional explicit 3D regularization or training overhead.

We propose a practical sampling framework to achieve the above-mentioned goal by steering the standard diffusion sampling trajectory with an energy term coupling two sampling trajectories. This method ensures that each sample from one diffusion model remains within its own distribution while being guided by the other. In particular, samples from the multi-view diffusion model maintain multi-view consistency while being steered by the content edits from the 2D model. Conversely, the 2D model is steered so that its edits remain faithful to the inputs while being consistent across independently edited frames.

Our solution is conceptually simple, broadly applicable, and adaptable to a variety of settings. We showcase its effectiveness across three distinct multi-view image editing tasks: multi-view spatial editing, stylization, and relighting. Through comprehensive experiments on each task, we demonstrate the advantages of our method over the state-of-the-art. We further validate the generalizability of our approach by applying it to diverse diffusion backbones and latent spaces, underscoring its promise as a general multi-view image editing engine.

## 2 RELATED WORK

**Test-time diffusion guidance.** Test-time guidance approaches for diffusion models have been proposed to steer diffusion models toward external objectives. Test-time scaling methods (Ma et al., 2025; Li et al., 2024), such as rejection sampling or verifier-based search over large latent spaces, passively filter generated samples. In contrast, optimization-based guidance actively steers diffusion trajectories, offering a more efficient alternative. A widely used technique is classifier guidance, where a discriminative classifier steers the diffusion trajectory toward a target label (Dhariwal & Nichol, 2021). When the objective is differentiable, gradient-based guidance can be directly applied during sampling (Bansal et al., 2024) In other cases, prior work has explored diffusion guidance using degradation operators, which require additional assumptions in the forward process, e.g., as

in linear inverse problems (Kawar et al., 2022; Wang et al., 2023a; Chung et al., 2023). However, in more general scenarios, such constraints are often intractable, making the proposed framework particularly suitable for these settings.

**3D and multiview editing.** With the advent of diffusion models capable of producing high-quality 2D image edits (Cao et al., 2023; Mokady et al., 2023), a natural question has been how to leverage those capabilities for 3D editing. One common approach is to optimize a 3D representation, such as Neural Radiance Fields (NeRF) (Mildenhall et al., 2020), so that its multiview renderings satisfy the editing goal. Bridging diffusion and NeRF can be achieved either by modifying the training dataset during the optimization loop (Haque et al., 2023; Wu et al., 2024a) or through score distillation sampling (Poole et al., 2023; Wang et al., 2023b; McAllister et al., 2024; Yan et al., 2025). However, both approaches are prone to visual artifacts, which is fundamentally caused by the fact that 2D diffusion models lack 3D consistency awareness. To address this fundamental challenge, prior work has directly trained multiview diffusion models (Litman et al., 2025; Alzayer et al., 2025a) for consistent editing. However, training a multiview diffusion model for each individual editing task is computationally expensive, and suitable training datasets are scarce. In our approach, we propose reusing existing multiview *generation* models (Gao et al., 2024; Zhou et al., 2025) for multiview *editing* by combining them with a 2D editing model, thereby incurring no additional training cost. In contrast to NeRF-based approaches, our method does not require a costly optimization process, as it relies solely on feed-forward sampling.

**Compositional diffusion sampling.** Compositional sampling methods for diffusion models have been proposed to combine the priors of multiple models. Examples include product-of-experts sampling (Hinton, 2002; Zhang et al., 2025b), which samples from the product distribution of individual models. However, this approach imposes a strict requirement that valid samples lie in the intersection of the support of each model and fails when no such joint support exists. MultiDiffusion (Bar-Tal et al., 2023) and SyncTweedies (Kim et al., 2024) apply score composition for stitching panoramas or large images. However, their primary focus is on handling out-of-distribution scenarios, such as oversized images, whereas our work emphasizes remaining within each model's prior distribution while steering generation toward satisfying cross-model constraints. Prior works Liu et al. (2022); Du et al. (2023) address inference-time composition for diffusion models, but these works focus on the same data modality. In contrast, our work bridges 2D and 3D modalities to tackle the practical challenge of 3D data sparsity.

## 3 METHOD

### 3.1 BACKGROUND

**Diffusion Models.** Let $x_0 \sim p_{\text{data}}(x_0)$ be a data sample and consider the forward noising process:

$$q(x_t \mid x_{t-1}) = \mathcal{N}(x_t \mid \sqrt{1 - \sigma_t} x_{t-1}, \sigma_t I), \tag{1}$$

with a variance schedule $\{\sigma_t\}_{t=1}^T$. (Ho et al., 2020) proposes to train a neural network $\epsilon_\theta(x_t, t)$, where $\theta$ denotes network parameters, such that when starting with an initial noise $x_T \sim \mathcal{N}(0, I)$, it allows one to gradually denoise the sample to $x_0 \sim p_{\text{data}}(x_0)$ via

$$\hat{x}_0 = \frac{1}{\sqrt{\bar{\alpha}_t}}(x_t - \sqrt{1 - \bar{\alpha}_t}\epsilon_\theta(x_t)) \tag{2}$$

$$x_{t-1} = \sqrt{\bar{\alpha}_{t-1}}\hat{x}_0 + \sqrt{1 - \bar{\alpha}_{t-1}}\epsilon_\theta(x_t) + \sigma_t z, \tag{3}$$

where $\alpha_t = 1 - \sigma_t, \bar{\alpha}_t := \prod_{s=1}^t \alpha_s$. The next-step prediction $x_{t-1}$ is obtained by computing the *clean* image estimate $\hat{x}_0$ and re-injecting a decreasing amount of random noise $z \sim \mathcal{N}(0, I)$.

### 3.2 COUPLED DDPM SAMPLING

**Problem.** Given two diffusion models $\epsilon_{\theta^A}$ and $\epsilon_{\theta^B}$ for a shared data domain $\mathbb{R}^d$ and with a shared DDPM schedule, our goal is to obtain two samples $x^A, x^B \in \mathbb{R}^d$ such that they follow the data distribution prescribed by the pre-trained models $p_{\text{data}}^A(x)$ and $p_{\text{data}}^B(x)$, respectively, while staying close to each other. This objective can be interpreted as tilting the distribution $p_{\text{data}}^A(x)$ to be close to

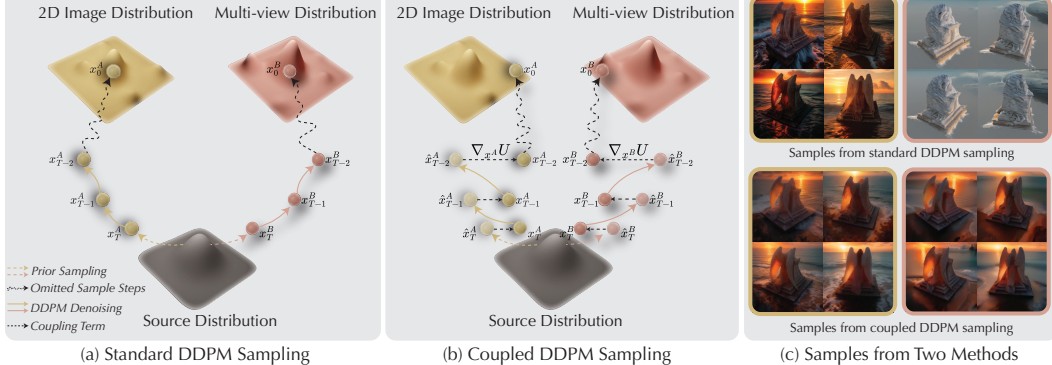

Figure 3: **Overview of the proposed coupled sampling method.** Given two target statistical distributions modeled with diffusion models: (a) standard DDPM sampling generates two instances independently, using scores from each distribution, which leads to samples without spatial alignment; (b) in contrast, the proposed coupled DDPM sampling introduces coupling terms $\nabla U$ that pull the two sample paths together, producing spatially and semantically aligned outputs; and (c) as illustrated, yellow-bordered images are drawn from the 2D image distribution, while pink-bordered images are drawn from the multi-view distribution.

---

**Algorithm 1** Coupled DDPM Sampling

---

1: $\theta_{2D}$: Text2Image diffusion model
2: $\theta_{MV}$: Text2MultiView diffusion model
3: $x_{T,2D}, x_{T,M} \sim \mathcal{N}(0, I)$: initial latents
4: $x_{T,2D}, x_{T,M}$ shapes: $N \times H \times W \times C$ where $N$ is # of views
5: **for** $t \in T, ..., 0$ **do**
6:     $\hat{x}_{t-1,2D}, \hat{x}_{2D,0} \leftarrow \text{Denoise}(\theta_{2D}, t, x_{t,2D})$               ▷ Diffusion model call to estimate x0
7:     $\hat{x}_{t-1,MV}, \hat{x}_{MV,0} \leftarrow \text{Denoise}(\theta_{MV}, t, x_{t,MV})$
8:     $x_{t-1,2D} \leftarrow x_{t,2D} - \lambda(\hat{x}_{2D,0} - \hat{x}_{MV,0})$
9:     $x_{t-1,MV} \leftarrow x_{t,MV} - \lambda(\hat{x}_{MV,0} - \hat{x}_{2d,0})$
10: **end for**

---

a sample $x^B(x) \sim p_{\text{data}}^B(x)$, and vice versa. We introduce a coupling function $U : \mathbb{R}^d \times \mathbb{R}^d \to \mathbb{R}$ that measures the closeness of two samples. A natural choice is the Euclidean Distance and in this work, we use $U(x, x') = -\frac{\lambda}{2}\|x - x'\|_2^2$ with a constant coefficient $\lambda \in \mathbb{R}$. Formally, our objective is written as

$$\min_{x^A, x^B} \mathcal{J}^A(x^A, x^B) + \mathcal{J}^B(x^A, x^B), \quad \text{where} \tag{4}$$

$$\mathcal{J}^A(x; x') := p_{\text{data}}^A(x) \exp U(x, \text{sg}(x')), \tag{5}$$

$$\mathcal{J}^B(x; x') := p_{\text{data}}^B(x) \exp U(\text{sg}(x), x'), \tag{6}$$

where sg denotes the stop gradient operation. Taking the gradients:

$$\nabla_x \mathcal{J}^i(x, x') = \nabla_x \log p^i(x) + \nabla_x U(x, x'), \quad i \in \{A, B\}. \tag{7}$$

Here, the additional term $\nabla_x U(x, x')$ biases the sample trajectory $\{x_t^i\}_t$ from the standard diffusion trajectory following $p^i(x)$ to satisfy the goal. Tilting diffusion model sampling towards inference-time reward functions or constraints has been widely studied for preference alignment (Wu et al., 2023) and inverse problems (Chung et al., 2023; 2022), with gradient likelihood of a form similar to Eq. (7), although typically under a fixed target. In contrast, in this work, the optimization target depends on another variable.

**Algorithm.** Let $x_t^A, x_t^B \in \mathbb{R}^d$ be two data samples.

$$x_{t-1}^A = \sqrt{\bar{\alpha}_{t-1}}\hat{x}_0^A + \sqrt{1 - \bar{\alpha}_{t-1}}(\epsilon_{\theta^A}(x_t^A) + \nabla_{\hat{x}_0^A} U(\hat{x}_0^A, \hat{x}_0^B)) + \sigma_t z^A, \quad z^A \sim \mathcal{N}(0, I), \tag{8}$$

$$x_{t-1}^B = \sqrt{\bar{\alpha}_{t-1}}\hat{x}_0^B + \sqrt{1 - \bar{\alpha}_{t-1}}(\epsilon_{\theta^B}(x_t^B) + \nabla_{\hat{x}_0^B} U(\hat{x}_0^B, \hat{x}_0^A)) + \sigma_t z^B, \quad z^B \sim \mathcal{N}(0, I). \tag{9}$$

Let $f^A(x_t^A; t) := \exp U(\hat{x}_0^A, \hat{x}_0^B) \propto \exp -\frac{1}{2}\frac{\|\hat{x}_0^A - \hat{x}_0^B\|_2^2}{1/\lambda} = \mathcal{N}(\hat{x}_0^B, 1/\sqrt{\lambda}I)$, providing the interpretation that $f^A(x_t^A; t)$ assigns low energy to $\hat{x}_0^A$ close to $\hat{x}_0^B$ in during the sampling process, and

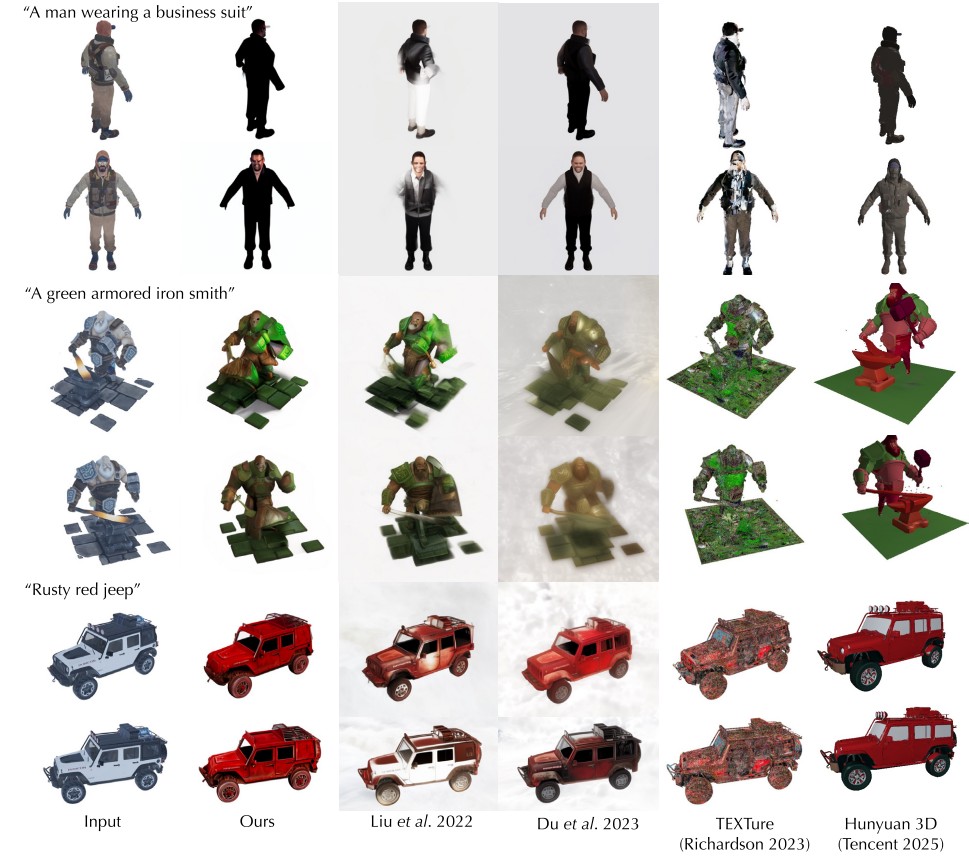

Figure 4: **Multi-view stylization.** We show three examples of multi-view stylization of our method against the baselines. Prior work on combining diffusion models (Liu et al., 2022; Du et al., 2023) suffer from inconsistencies across frames. SDS based methods (Richardson et al., 2023) suffer from severe artifacts. Hunyuan 3D's results follow the prompt loosely when doing retexturing.

similarly for $x_t^B$. This term effectively serves as a soft regularization that encourages two samples to stay close. The gradient term $\nabla_x U(x, x') = -\lambda(x - x')$ is easy to compute with minimal computation overhead. The sampling algorithm is summarized in Algorithm 1.

## 4 EXPERIMENTS

*We refer the readers to the supplementary webpage for video results.*

To demonstrate the versatility of our method, we select tasks that highlight various editing aspects.
1) *Spatial editing*: We use Magic Fixup (Alzayer et al., 2025b) to highlight the ability of making geometric changes in a scene. 2) *Stylization*: We perform stylization using Control-Net (Zhang et al., 2023) with edge control, demonstrating how we can alter the general appearance of the input while preserving its overall shape. 3) *Relighting*: We perform relighting using two different models: 1) Neural-Gaffer (Jin et al., 2024), which takes an explicit environment map as input, and 2) IC-Light (Zhang et al., 2025a), which is text-conditioned, producing more diverse edits.

For each of these tasks, we begin with a collection of input images and additional task-specific conditioning. The 2D model is capable of editing each image individually, but this often leads to inconsistencies across the set. In contrast, the multi-view model (Zhou et al., 2025) is a novel view synthesis model that takes a set of consistent images and generates novel views. Our pipeline first edits a single image using the 2D model and then uses it as a reference for the multi-view model. However, editing only a single image is insufficient to fully preserve the identity of the input, as illustrated in Figure 2. To address this, we couple the two models, enabling the multi-view model to maintain identity while ensuring consistency across multiple views. We perform the coupling in the latent space, and in all these experiments, both the image editing models and the multi-view model operate in the latent space of Stable Diffusion 2.1 (Rombach et al., 2022).

For each task, we adopt Liu et al. (2022) and Du et al. (2023) as general-purpose baselines for combining our two diffusion models. We also include task-specific baselines tailored to each scenario.

Table 1: Quantitative comparison on spatial editing. We evaluate against GT renders of the target edit, and use MEt3r for geometric consistency.

| Method | Per-image metrics | | | MV metric | |
|---|---|---|---|---|---|
| | PSNR ↑ | SSIM ↑ | LPIPS ↓ | MEt3r ↓ | Users ↑ |
| Per-image | 16.5 | 0.550 | 0.253 | 0.353 | - |
| Image-to-MV | 12.84 | 0.400 | 0.556 | 0.417 | - |
| Liu et al. (2022) | 16.5 | 0.530 | **0.354** | 0.368 | 9% |
| Du et al. (2023) | 16.7 | 0.548 | 0.411 | 0.344 | 1% |
| SDEdit | 15.4 | 0.458 | 0.468 | 0.393 | 11% |
| **Ours** | **17.0** | **0.550** | 0.421 | **0.335** | **80%** |

Table 2: Quantitative comparison on relighting. We evaluate against GT relighting results in terms of per-image metrics, and evaluate multi-view consistency with MEt3r.

| Method | Per-image metrics | | | MV metric | |
|---|---|---|---|---|---|
| | PSNR ↑ | SSIM ↑ | LPIPS ↓ | MEt3r ↓ | Users ↑ |
| Per-image | 22.7 | 0.862 | 0.159 | 0.243 | - |
| Image-to-MV | 19.3 | 0.815 | 0.193 | 0.229 | - |
| Liu et al. (2022) | **23.2** | **0.871** | **0.152** | 0.220 | 10% |
| Du et al. (2023) | 22.1 | 0.863 | 0.158 | **0.217** | 19% |
| NeRF + NG | 22.4 | 0.865 | 0.162 | **0.217** | 25% |
| **Ours** | **23.2** | 0.868 | 0.157 | **0.217** | **46%** |

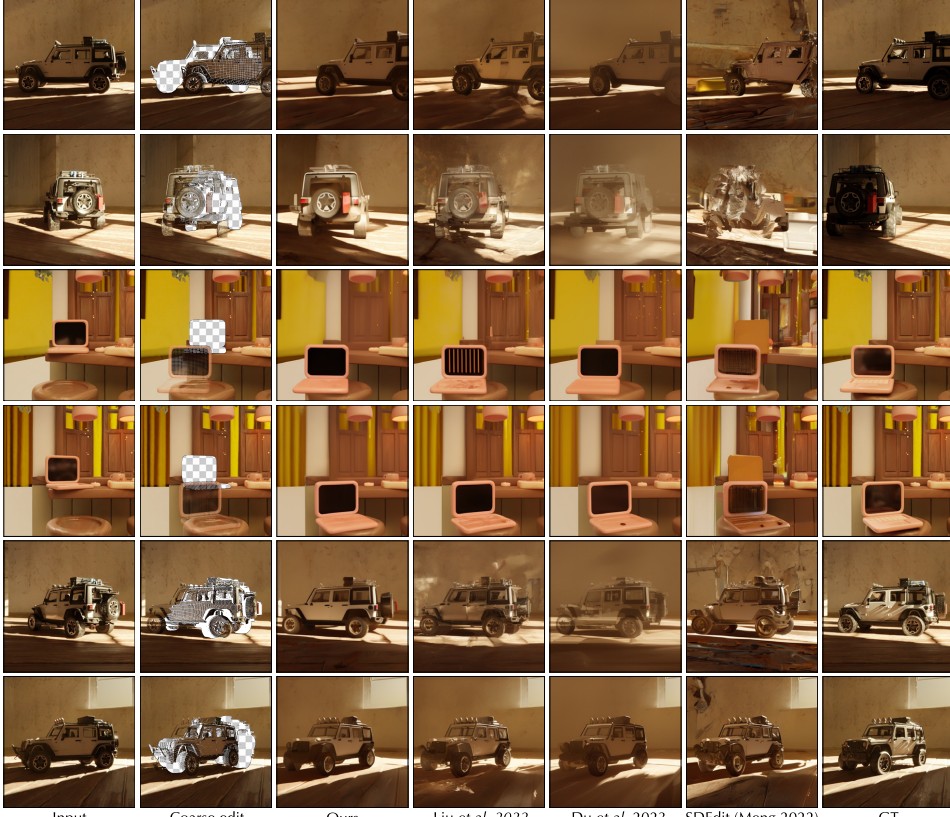

Input     Coarse edit     Ours     Liu *et al. 2022*     Du *et al.* 2023     SDEdit (Meng 2022)     GT

Figure 5: **Qualitative comparison on multi-view spatial editing.** The baselines struggle in preserving the identity of the input, and produce flickering artifacts across edited frames, while our results achieve both editing targets and multi-view consistency.

To provide a comprehensive evaluation, we conduct user studies with 25 participants for all tasks, comparing our approach to all baselines using best-of-$n$ preference questions.

## 4.1 MULTI-VIEW SPATIAL EDITING

Spatial editing is challenging because it requires accurately harmonizing the scene, including object interactions and changes in shadows and reflections resulting from edits. There are no large-scale datasets available for training spatial editing models. As a result, previous work on 2D spatial editing has relied on large-scale video datasets (Wu et al., 2024b; Cheng et al., 2025; Alzayer et al., 2025b) to learn natural object motion. However, such data sources do not exist for multi-view datasets, as dynamic multi-view or 4D datasets are extremely scarce and are typically created only for evaluation purposes. Our coupled sampling paradigm addresses this gap.

We use Magic Fixup (Alzayer et al., 2025b) for the 2D editing model. This model takes the original image and a coarse edit that specifies the desired spatial changes. For multi-view editing, it is necessary to apply the edit consistently across all views. In our experiments, we unproject the target object in each image using a depth map. We then apply a 3D transformation to the object and reproject it into the image. As a baseline, we also use SDEdit (Meng et al., 2022), which similarly accepts a coarse edit. Figure 5 presents three different coarse edits, with two frames from each edit

Table 3: Quantitative comparison on stylization. We evaluate the temporal and subject consistency, and MEt3r score for geometric consistency. CLIP score is computed against the edit prompt.

| | Per-img metric | MV metrics | | | | |
|---|---|---|---|---|---|---|
| Method | CLIP score ↑ | Temp. consis. ↑ | Subject consist. ↑ | MEt3r ↓ | User pref. ↑ | Mesh-free |
| Per-image (Zhang et al., 2023) | 30.0 | 0.922 | 0.740 | 0.546 | - | ✓ |
| Image-to-MV (Zhou et al., 2025) | 29.5 | 0.927 | 0.787 | 0.382 | - | ✓ |
| TEXTure (Richardson et al., 2023) | 28.4 | 0.967 | 0.748 | 0.426 | 14% | X |
| Hunyuan3D (Team, 2025) | 29.9 | 0.952 | 0.754 | 0.391 | 8% | X |
| Liu et al. (2022) | 30.1 | 0.934 | 0.759 | 0.461 | 19% | ✓ |
| Du et al. (2023) | **30.2** | 0.926 | 0.762 | 0.461 | 12% | ✓ |
| **Coupled Sampling (Ours)** | 29.68 | **0.946** | **0.807** | **0.392** | **47%** | ✓ |

shown to illustrate consistency. In the first example, we find that our method correctly translated and rotated the car, while preserving the identity of the input. By contrast, the baselines struggle to maintain the back view of the scene. In the final edit, our method produces smooth shadows that match the ground truth, whereas the baseline results in highly irregular shadows.

To quantitatively evaluate performance, we render the ground truth 3D transformation for each edit using Blender. We use standard reconstruction metrics, and MEt3r (Asim et al., 2025), which measures the 3D consistency of multi-view outputs. Table 1 demonstrates that our method achieves higher PSNR and SSIM scores, along with superior multi-view consistency.

## 4.2 MULTI-VIEW STYLIZATION

Stylization is a common application of diffusion models, where an input sequence, the spatial structure of the desired output, and a text prompt specifying the style are provided. Control-Net (Zhang et al., 2023) enables this type of stylization by incorporating geometry-related conditioning, such as the Canny edges of an image. Because ControlNet is trained on a large dataset, it achieves higher text fidelity than text-to-MV models. A closely related task is 3D re-texturing, in which a 3D mesh is given and a new texture is generated using a generative model. To assess our method, we rendered ten different scenes and applied stylization to each using user-defined prompts. For a comprehensive comparison, we also include baselines that operate directly on the 3D mesh, such as TEXTure (Richardson et al., 2023), which synthesizes new textures using SDS (Poole et al., 2023), and Hunyuan3D (Team, 2025), which employs a feed-forward multi-view model to generate textures. We omit InstructNeRF2NeRF as it fails on our inputs. In Fig. 4, we present results from three representative examples. In the first example, score averaging methods have difficulty preserving the identity of the edited subject, resulting in color changes or the changing identity across frames. In contrast, TEXTure exhibits severe artifacts due to its SDS-based approach. Hunyuan3D produces very simple edits that often do not align with the text prompt.

Although the quantitative evaluation of stylization remains challenging, we assess both temporal and subject consistency in our generated videos using VBench Zhang et al. (2024) and measure geometric consistency with MEt3r (Asim et al., 2025). Our results show that our method achieves superior temporal and subject consistency compared to previous approaches for combining diffusion models. For reference, we also report results from mesh-based methods on rendered videos, which are inherently temporally consistent due to the underlying mesh representation.

## 4.3 MULTI-VIEW RELIGHTING

**Environment map conditioned relighting.** When the variance of the 2D diffusion results is low, meaning the sampling distribution is narrow, radiance fields can effectively regularize inconsistencies. However, this requires obtaining a consistent geometry beforehand. As an alternative, we demonstrate that a multi-view diffusion model can regularize inconsistencies in 2D relighting through coupled sampling. Figure 6 presents two relighting examples to illustrate this. We observe that prior methods for combining diffusion models (Liu et al., 2022; Du et al., 2023) can introduce flickering artifacts, as evidenced by abrupt color changes in the top two rows. In contrast, NeRF-based approaches may incorrectly attribute lighting variance to view-dependent effects, as illustrated in the bottom two rows of the backpack example. To quantitatively compare these methods, we use the 3D objects from Neural-Gaffer (Jin et al., 2024), and add both a diffuse and a glossy object, resulting in a total of seven objects with five relightings each. We compute per-image reconstruction metrics and geometric consistency using MEt3r, as shown in Table 2. Although these metrics do not capture subtle lighting flicker, our method achieves competitive results in both reconstruction

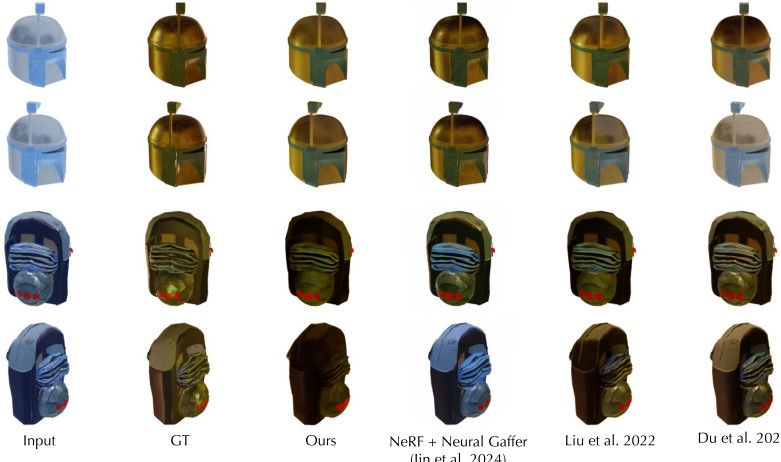

Figure 6: **Qualitative comparison on environment map based relighting.** Other methods tend to produce flickering artifcats (notice the change in color in the first two rows for Liu et al. (2022); Du et al. (2023)). The usage of NeRF will make the lighting changes to be baked into the view dependent effects. Our method achieves the best overall result.

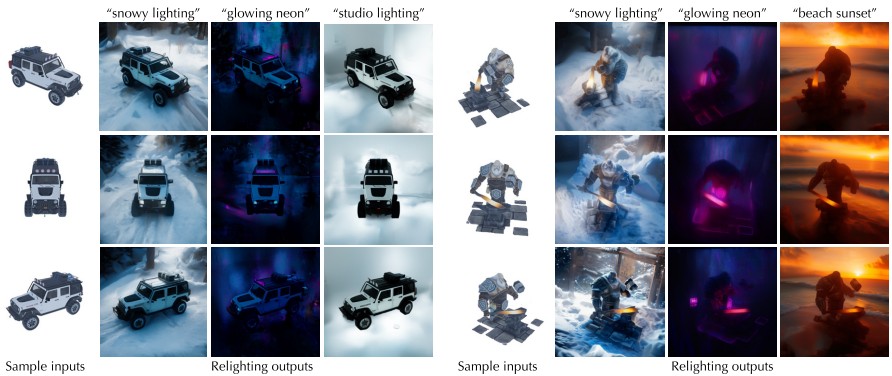

Figure 7: **Text based relighting.** We combine IC-Light (Zhang et al., 2025a), which enables text-based relighting with stable virtual camera to obtain multi-view results.

and consistency. Importantly, we also report metrics for relighting each image individually, which serves as a coarse upper bound, and observe no degradation in performance.

**Text conditioned relighting.** To show more drastic relighting outputs, we use IC-Light (Zhang et al., 2025a), which operates by relighting the object and adding a suitable background. While Stable-Virtual-Camera (Zhou et al., 2025) may have a weak prior for regularizing backgrounds due to its training data, we find that it still ensures the object is consistently lit across frames. In Fig. 7 we show diverse multi-view relighting results using our method.

## 5 ANALYSIS EXPERIMENTS

In this section, we demonstrate that the benefits of coupled sampling extend to various models, and analyze how varying the guidance strength in our approach influence the results.

**Backbone variations.** In Section 4, we presented multi-view editing results using Stable Virtual Camera (Zhou et al., 2025). Here, we further examine the impact of coupling on text-to-multi-view models, specifically MVDream (Shi et al., 2024), which extends Stable Diffusion 1.5 to produce four consistent views, and MV-Adapter (Huang et al., 2024), which leverages the more advanced SDXL backbone and operates in the SDXL latent space. For coupling, we use SD1.5 and SDXL as the respective text-to-image models. As shown in Figure 8, text-to-multi-view models often generate objects with a CGI-like appearance, likely due to their training on datasets such as Objaverse (Deitke et al., 2023). Introducing our coupling approach encourages the multi-view samples to better resemble real images, as modeled by the 2D diffusion models.

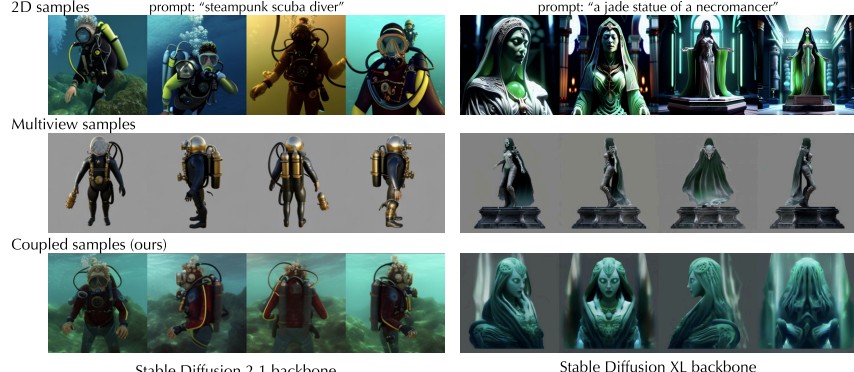

Figure 8: **Coupling in different mutli-view models.** We implement coupling on T2I and T2MV models with two different backbones. We couple SD2.1 with MVDream (Shi et al., 2024), and SDXL with MVAdapter (Huang et al., 2024) which operates in SDXL latent space. In both cases, the coupled multiview samples show an increase in realism and decrease in "objaverse" appearance.

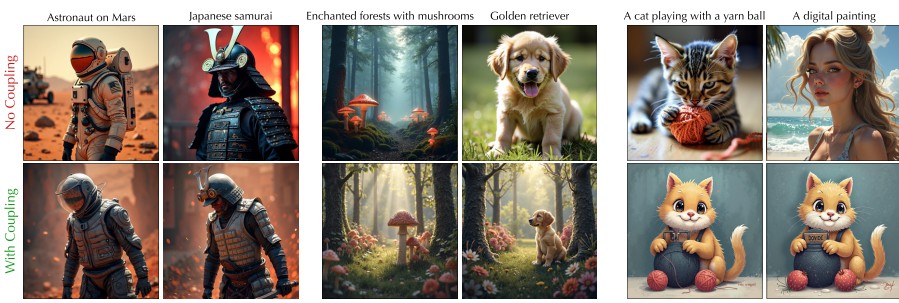

Figure 9: **Image space coupling.** Using Flux, we perform coupled sampling on different prompts. We show that the coupled samples are spatially aligned while being faithful to the prompt.

**Coupling Text-to-image flow models.** Coupled diffusion sampling can be applied to both 2D and multi-view settings. To illustrate the effects of coupled sampling, we implement our method using the text-to-image model Flux (Labs, 2024). Although Flux is a flow-based model (Lipman et al., 2023; Liu et al., 2023), we show that our coupling approach remains effective. We test coupled sampling by generating two samples from the same model, each conditioned on a different prompt. As shown in Fig. 9, without coupling, the outputs are typically very distinct. With the coupled sampling, the outputs become spatially aligned while still reflecting their respective prompts.

**Guidance strength analysis.** We quantitatively evaluate the effects of guidance strength $\lambda$ on spatial editing performance. When $\lambda$ is very small, the model output resembles image-to-MV sampling, resulting in low reconstruction performance. As $\lambda$ increases, reconstruction performance improves. However, with further increases in $\lambda$, consistency across frames degrades as the outputs become more similar to 2D model samples and eventually collapse.

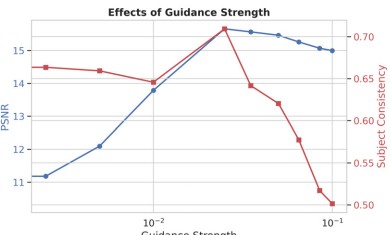

Figure 10: **Guidance strength analysis.** As we increase the guidance strength, the reconstruction improves but the consistency drops.

## 6 DISCUSSION AND CONCLUSIONS

We introduce a simple and effective approach for coupling diffusion models, enabling 2D diffusion models to generate consistent multi-view edits when used with multi-view diffusion models. Our method is efficient, versatile, and achieves high-quality results. By guiding the diffusion sampling process, our approach produces outputs that retain the strengths of the underlying models, while also inheriting their limitations.

We believe this coupling strategy has potential applications beyond multi-view editing. In the future, our paradigm could extend the capabilities of image-editing models to video editing by integrating with video diffusion models, without incurring additional computational overhead.

**Reproducibility statement.** A key advantage of our approach is its simplicity: it requires only a straightforward modification to the standard DDPM sampling procedure, which can be readily implemented based on the details provided in this paper. To promote reproducibility and facilitate further research, we will publicly release our coupled sampling code upon acceptance.

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

## A  ADDITIONAL DISCUSSION OF LIMITATIONS

While the proposed method offers a simple and efficient framework for multi-view consistent image editing, several notable limitations remain. First, running both the 2D editing model and the multi-view diffusion model in parallel increases memory and computational requirements. This limitation could be addressed in future work by exploring adaptive guidance strength or applying guidance during only a subset of sampling steps. Second, the edited outputs are not perfectly 3D consistent

compared to test-time optimization-based methods. This residual inconsistency can be further reduced by robustly fitting a NeRF or 3D Gaussian Splatting model to the generated views, as shown in prior work Haque et al. (2023); Weber et al. (2024), while still maintaining much faster inference than optimization-based approaches.

## B Implementation details

As our primary multi-view generation base model, we use Stable-Virtual-Camera (SVC) which is trained to process 21 frames at once, and use one or several consistent images for novel view synthesis. As we would like to edit a collection of views, we do not have access to more than one consistent *edited* photos, since we can only edit one image at a time with the 2D editing model. In our experiments, we edit a reference image, and then use it as the conditioning view. Note that this conditioning view has great influence on the outcome, as it dictates the distribution of acceptable 3D scenes that SVC would synthesize.

We transform stable-virtual-camera into DDPM by converting the EDM based sampler into a DDPM scheduled by computing converting the noise levels into the appropriate alphas. Afterwards, since SVC was trained with a shifted noise schedule compared to SD2.1 image models, we re-align SVC's schedule with the 2D model's schedule for the coupled sampling to be effective.

We conduct our experiments using NVIDIA A6000 GPUs. As our approach only requires a feed forward pass, the memory requirement is equivalent to the combined memory of the two models used. A better memory utilization can be further achieved by loading and off-loading the models from the GPU, as we can run them sequentially and then compute the coupling term. We use 50 denoising steps for spatial editing and stylization, and 100 denoising steps with Neural Gaffer relighting. The runtime of the sampling process is 130 seconds using our GPU resources for generating the full 21 frames sequence.

In the experiments with Neural-Gaffer, one challenge is that Neural-Gaffer is trained on 256x256 images. On the other hand, SVC was trained on 576x576 images. We found that SVC performs very poorly on images of that size, and neural-gaffer does not generalize to 512x512 images or larger. After experimenting with the models, we found that at resolution size of 384x384, both models perform reasonably well and adopt that for the neural-gaffer experiments.

## C Coupled Diffusion Sampling with Flow Models

Note that Flux (Labs, 2024), the text-to-image model used in Sec. 5 is a flow model. To sample from Flux using our proposed sampling method, first we transform the velocity $v_\theta(x_t)$ to the score function $s_\theta(x_t)$, as it can be linearly transformed into score functions via $s_\theta(x_t) = -\frac{-tv_\theta(x_t)+x_t}{1-t}$. Then transform the inference schedule to be DDPM via time reparameterization (Lipman et al., 2024) by computing the appropriate alpha values that match the noise levels associated with each time step.

### C.1 Effects of Guidance Strength

As an additional illustration, in Fig. 11 we show how the samples change as we increase the coupling strength, while using the same initial random noise and randomness seed.

## D User study on IC-Light

For completion, we include the results of our user study on IC-Light in Tab. 4. We show that our outputs are preferred by users over either of prior work on combinign diffusion models, as they tend to produce high flickering artifacts in the relit outputs.

## E Effects of stochasticity

One observation one would make is that our coupling term resembles linearly combining the intermediate samples of the two models, so one may wonder why we do not simply get images that are

Prompt: Japanese Samurai

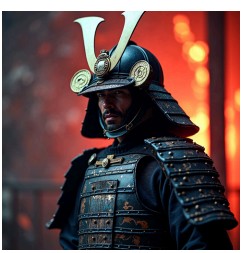 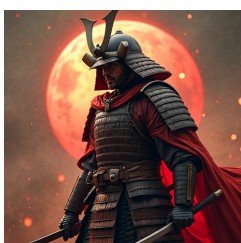 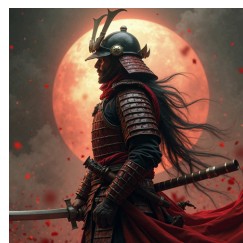 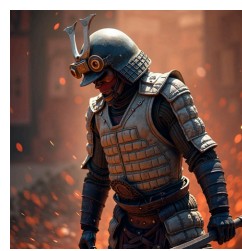

Coupling Strength 0.0      Coupling Strength 0.0025      Coupling Strength 0.005      Coupling Strength 0.01

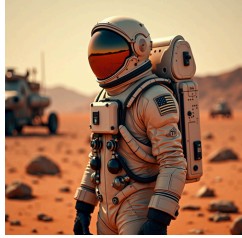 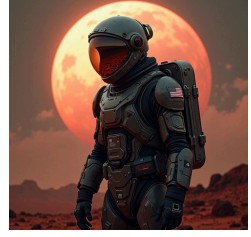 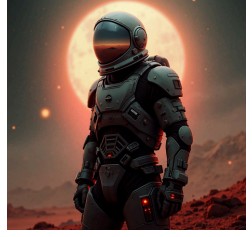 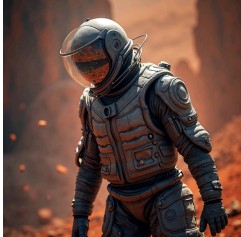

Prompt: Astronaut on Mars

Figure 11: **Effects of coupling strength.** We illustrate the effects of coupling strength on the spatial alignment between samples as we vary the coupling strength while keeping the initial noise and random seed.

Table 4: User study results on text-based relighting.

| Method | User pref. ↑ |
|---|---|
| Liu et al. (2022) | 24% |
| Du et al. (2023) | 26.5% |
| **Coupled Sampling (Ours)** | **49.5%** |

a linear average of the two outputs. Indeed, when we use a deterministic sampler, like Euler Discrete Sampler that's commonly used, this is the outcome that we encounter as we show in Fig. 12. However, when using a stochastic sampler like DDPM where noise is injected at every timestep, the model needs to correct for the added noise. When we include our coupling term in the stochastic step, the model can naturally correct or reject parts of the guidance that steers it away from its training distribution. This is also the reason we make our coupling term to be correlated with the noise level, by scaling it with $\sqrt{1-\alpha_t}$, since at step $t$, the model has the ability to correct for noise at that level, but steering the sample by a larger magnitude risks pulling the intermediate latents outside of the training distribution. Additionally, as intuitively understood about diffusion sampling, at the later time steps the structure of the outputs is already determined, so shifting the intermediate latents in a large direction can disrupt the sampling process.

## F    ADDITIONAL T2MV RESULTS

In Fig. 13 and Fig. 14, we highlight additional results from coupling text-to-multi-view models along with text-to-image models.

## G    APPLICATIONS WITH MV-ADAPTER

One of the limitations of MV-Adapter is that it can only generate fixed set of camera views, making its utility for editing limited. Nonetheless, we show that we can still use it by editing the outputs it

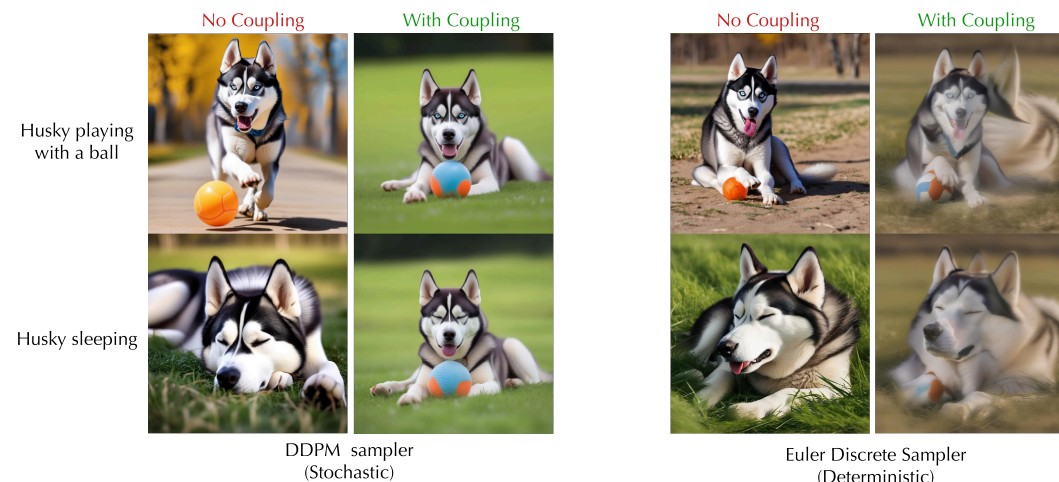

Figure 12: **Sampler comparison.** When using a stochastic sampler, the coupling can lead to natural guidance pulling the outputs towards each other. On the other hand, a deterministic sampler would simply output the average of both samples, as ODE based sampling does cannot recover from noisy guidance.

produces by performing coupling with single-image editing models. In Fig. 15, we show an example of using MV-Adapter for stylization, and relighting.

## H OUTPUTS OF INSTRUCTNERF2NERF

When running InstructNeRF2NeRF on our input sequences used for stylization with the same number of frames as our method and other baselines (21 frames), we find that the radiance field completely collapses. This is likely due to NeRF's inability to gradually handle inconsistency with less dense camera coverage.

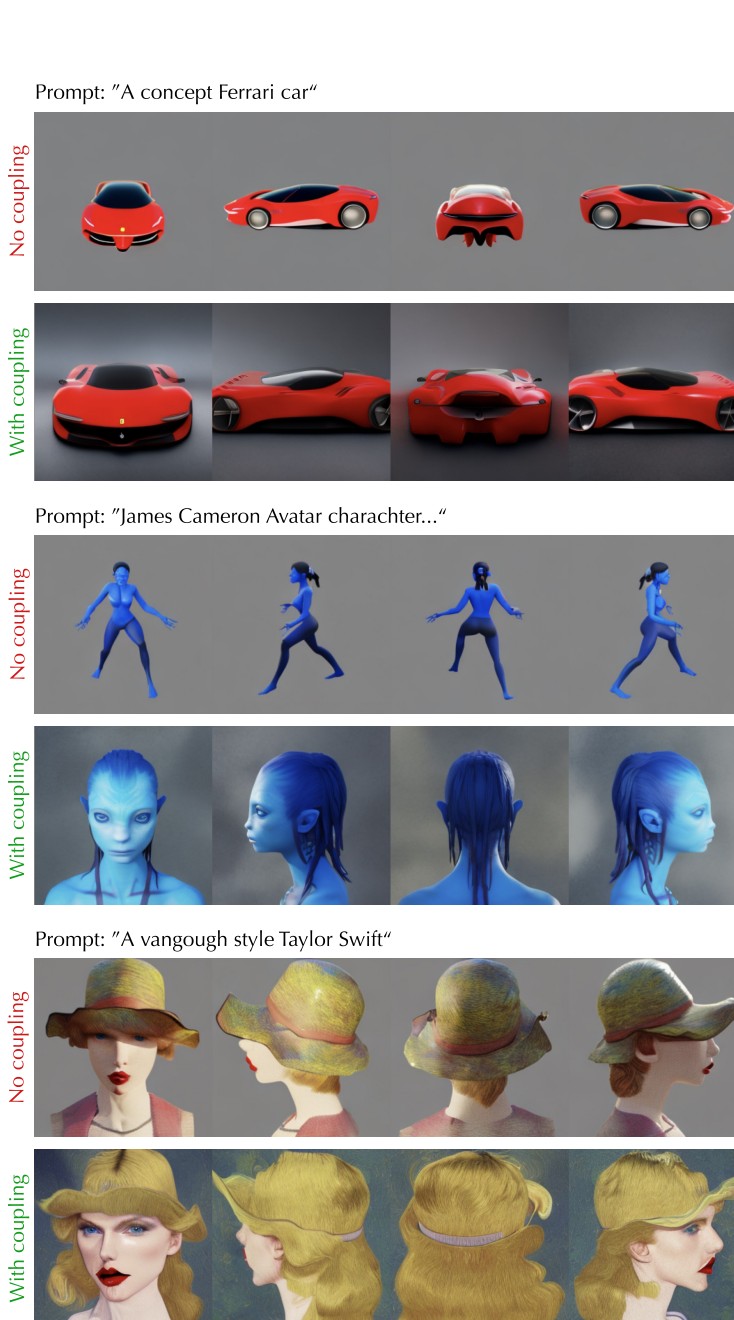

Figure 13: **Additional MVDream T2MV coupling results.** Here we show additional results on the output of Text-to-Multiview MVDream when coupled with Text-to-Image SD2.1.

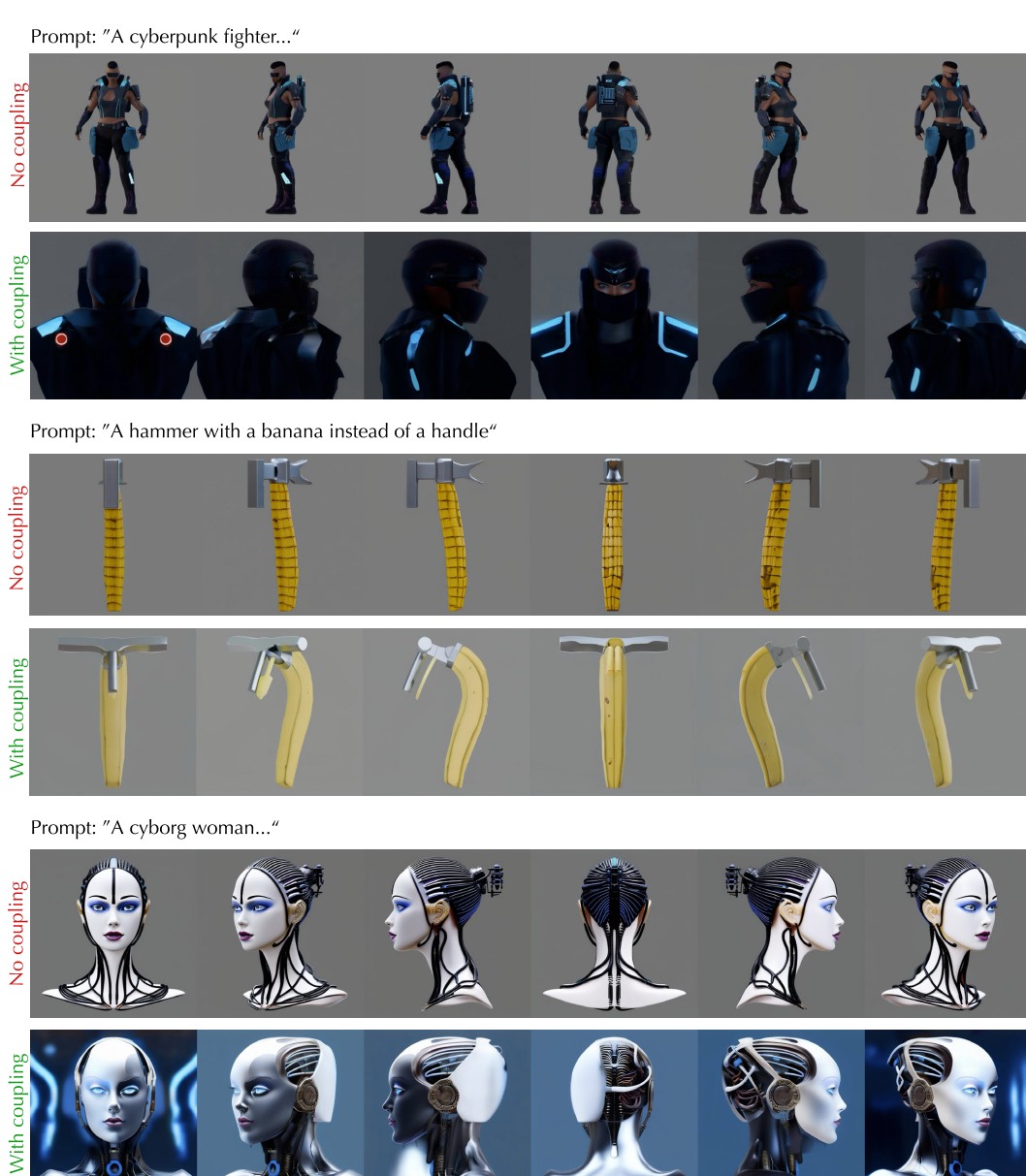

Figure 14: **Additional MV-Adapter T2MV coupling results.** Here we show additional results on the output of Text-to-Multiview MV-Adapter when coupled with Text-to-Image SDXL.

Input views

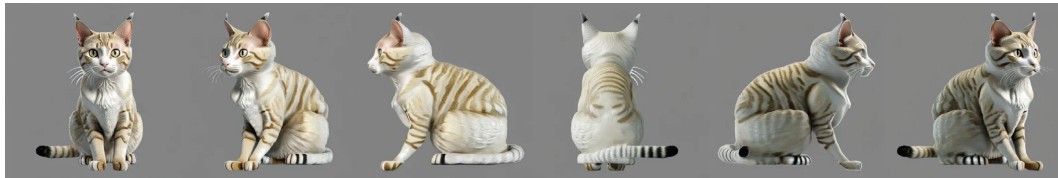

Stylization (MV-Adapter + Control-Net)

relighting prompt: a snow white cat

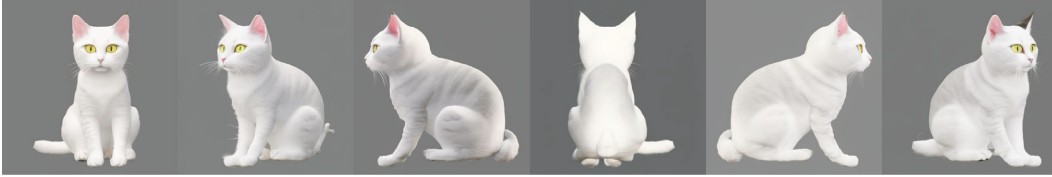

Relighting (MV-Adapter + IC-Light)

relighting prompt: a white and orange tabby cat in studio lighting

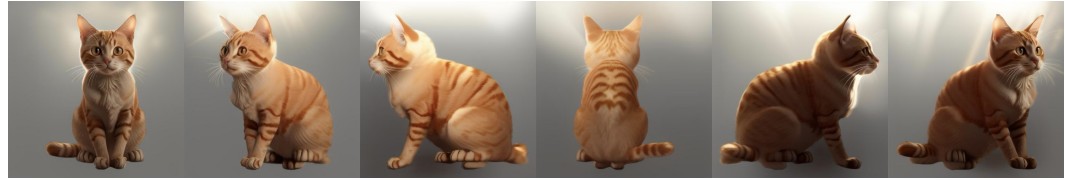

Figure 15: **Multiview editing with MV-Adapter.** Here we show editing results with MV-Adapter to achieve stylization by combining it with Control-Net (Zhang et al., 2023) and relighting using IC-Light (Zhang et al., 2025a).

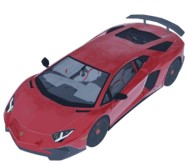
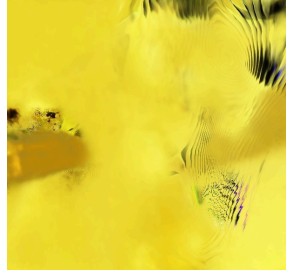
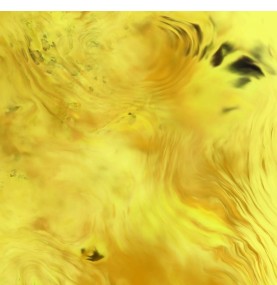

Sample input frame                     InstructNeRF2NeRF renders
+
Prompt: "make it a golden lamborghini"

Figure 16: **InstructNeRF2NeRF outputs.** When running InstructNeRF2NeRF (Haque et al., 2023) on our input views, we find that the editing training loop with InstructPix2Pix completely collapses.

