# OpenReview forum: "Coupled Diffusion Sampling for Training-Free Multi-View Image Editing"
_ICLR.cc/2026/Conference — ICLR 2026 Conference Withdrawn Submission_

### Official Review · Reviewer_yapA · 2025-10-26

**Soundness:** 3
**Presentation:** 3
**Contribution:** 3
**Rating:** 4
**Confidence:** 4

**Summary:**

This paper introduces "Coupled Diffusion Sampling," an inference-time method for achieving multi-view consistent image editing using pre-trained 2D editing and multi-view diffusion models. It concurrently samples two diffusion trajectories, one for 2D edits and another for multi-view consistency, using a simple coupling term to mutually guide them. This training-free approach avoids explicit 3D optimization, making it efficient and applicable across various editing tasks like spatial editing, stylization, and relighting.

**Strengths:**

The proposed Coupled Diffusion Sampling offers a computationally efficient and training-free solution for multi-view consistent image editing, elegantly sidestepping the lengthy optimization and instability issues common with explicit 3D representations. Its core strength lies in its ability to leverage existing pre-trained 2D and multi-view diffusion models by introducing a simple, inference-time coupling term, thereby achieving implicit 3D regularization. This approach demonstrates broad applicability across various editing tasks, including spatial editing, stylization, and relighting, and compatibility with diverse diffusion backbones, positioning it as a versatile and general framework for consistent multi-view content generation.

**Weaknesses:**

W1. The proposed method relies on a strict constraint that the two diffusion models must share the same data domain and diffusion schedule. This considerably limits the general applicability of the method. Although the authors emphasize that their approach is “broadly applicable” and works in a “variety of settings”, such requirements contradict these claims.

W2. A core component of the proposed method is the additional term $\nabla_x U(x, x’) = -\lambda (x - x’)$. Since this is governed by the hyperparameter $\lambda$, the performance of the model may highly depend on carefully tuning $\lambda$. This results in the method being overly reliant on heuristic engineering rather than principled modeling. The paper does not provide sufficient experiments or in-depth analysis on how $\lambda$ influences performance, except for a brief mention in Figure 10. Furthermore, it appears likely that $\lambda$ must be adjusted differently across datasets or input domains used in various tasks. If the success of the method depends heavily on such manual tuning, it could indicate that the approach is inherently unstable and lacks robustness.

W3. Missing evaluation protocol.
The paper lacks a clear evaluation protocol. It is not specified which datasets are used, how many instances per dataset are evaluated.

W4. Inadequate and misaligned baselines.
The paper compares against “Composable Diffusion(Liu et al, 2022)” and “Reduce, Reuse, Recycle (Du et al, 2022)” which are 2D compositional methods without 3D priors and are not trained for 3D content generation or editing. These choices are misaligned with the paper’s tasks. Strong, task-specific baselines are required.

• Stylization / 3D object editing: compare against 3D editing or 3D-aware generation methods such as Vox-E[1], Tailor3D[2], and Treliis[3].

• Spatial editing (3D-aware image edits): compare against Diffusion Handles[4], Magic Fixup[5], and GeoDiffuser[6].

W5. Limited editing diversity.
Most results only demonstrate texture- or color-level modifications, which are relatively simple forms of editing. To convincingly validate that the method supports general and versatile editing, it is necessary to include more challenging tasks such as object insertion, removal, or structural manipulation. Without these results, the method’s editing capability appears limited in scope.


---
[1] Sella, Etai, et al. "Vox-e: Text-guided voxel editing of 3d objects."

[2] Qi, Zhangyang, et al. "Tailor3d: Customized 3d assets editing and generation with dual-side images."

[3] Xiang, Jianfeng, et al. "Structured 3d latents for scalable and versatile 3d generation."

[4] Pandey, Karran, et al. "Diffusion handles enabling 3d edits for diffusion models by lifting activations to 3d."

[5] Alzayer, Hadi, et al. "Magic fixup: Streamlining photo editing by watching dynamic videos."

[6] Sajnani, Rahul, et al. "Geodiffuser: Geometry-based image editing with diffusion models."

**Questions:**

Q1. Does the hyperparameter $\lambda $remain fixed across all datasets and input images, or is it adjusted differently depending on the experiment?

Q2. In Equation (4), the objective is stated as a **min** problem for $J^A(x_A, x_B) + J^B(x_A, x_B)$. Given that $J^A$ and $J^B$ are defined as: $p_{\text{data}}(x) \exp U(x, x')$ where both $p_{\text{data}}(x)\) and \(\exp U(x, x')$ (with $U(x, x')$ being a negative Euclidean distance) are terms that represent *desirability* and should ideally be maximized, it seems that a **max** optimization would be more appropriate. Could the authors clarify this choice of **min** versus **max** in their formulation?

---

> ### Author Response · Authors · 2025-11-14
> **Thanks for the review**
>
> Thanks for the detailed review.
>
> - We would like to note that we include strong task-specific baselines for each task, including baselines that access privileged inputs like GT mesh and GT NeRF. For spatial editing, our "per image" baseline is using Magic Fixup, which is the 2D editing model we rely on for spatial editing.
> - The coupling strength coefficient $\lambda$ acts as the guidance strength, similar to prior test-time guidance methods. When setting it to 0, we obtain standard DDPM sampling, and as we increase it, the output samples start to follow the guidance objective. However, as in prior guidance techniques, setting the guidance too large can cause the samples to eventually collapse and produce artifacts. We will visually illustrate those artifacts in the revision.
>
> Thanks for the constructive review, and we will address the concerns in the revision

---

### Official Review · Reviewer_Hyz4 · 2025-10-26

**Soundness:** 3
**Presentation:** 3
**Contribution:** 2
**Rating:** 4
**Confidence:** 5

**Summary:**

This paper introduces a new method to edit a single-view but apply a multiview diffusion model to lift it to novel view images. The key idea is to combine the denoising direction of two denoisers (i.e., the single-view editing and the multiview diffusion). Experimental results on several examples show the proposed method enables multiview editing, relighting, and stylization.

**Strengths:**

The idea of combining the denoising directions of two diffusion models is interesting and may be useful in some cases.

**Weaknesses:**

1. The task is not well motivated. I'm not sure why we need to do multiview editing when only a single-view image is given. An alternative baseline would be we first edit the single-view image and then run the multiview diffusion model on the edited single-view image. In this case, we could easily get multiview-consistent edited results.
2. The quality is not good enough. The videos provided in the supplementary material clearly show inconsistent textures or geometry.
3. Some recent and strongly relevant baselines like Vox-E (ICCV'23), TIP-Editor (SIGGRAPH'24), and CMD (SIGGRAPH'25) are not discussed and compared. According to my experience, these trained model with explicit 3D representations could have much better performance on this task.

**Questions:**

A major question is why $U$ can be defined as an L2 distance. Here, a noticeable thing is that we apply the direction difference between two denoisers on the input view to all other novel views. I'm not sure whether the direction difference on the first view here is valid for all other novel views if they have different visual structures.

---

> ### Author Response · Authors · 2025-11-14
> **Thanks for the review**
>
> Thanks for the thoughtful review. We would like to clarify that our task takes a multi-view input (not a single view), and the goal is to edit all the views consistently with each other.
> We do include the baseline of editing a single image, and generating additional views (see Fig. 2 and supplementary webpage with the baseline called "Image-to-MV"). Since this is under-constrained, the additional views are hallucinated and do not match the unseen input views. Our proposed method allows the 2D editing model to steer the multi-view generation model during inference, effectively enforcing additional constraints to ensure that the output is faithful to the input images, while remaining multi-view consistent.
> We will include the references of suggested baselines in the revision. Note that for each task, we do use strong task-specific baselines, including baselines that have access to GT mesh and GT NeRF.
>
> Thanks again for the constructive review, and we will revise our paper accordingly.

---

### Official Review · Reviewer_gCC9 · 2025-10-30

**Soundness:** 3
**Presentation:** 3
**Contribution:** 3
**Rating:** 4
**Confidence:** 4

**Summary:**

The authors present a novel inference-time method for multi-view consistent image editing using pre-trained 2D image editing models.
The main contribution lies in the coupled diffusion sampling which is a simple diffusion sampling technique that jointly samples two trajectories from both a multi-view image distribution and a 2D edited image distribution using a coupling term to enforce consistency.
This avoids expensive and time-consuming optimization common in NeRF models. The method proves effective and versatile over a range of tasks: spatial editing, stylization, and relighting.

**Strengths:**

1. Novel inference-time method: the method provdies an effective solution to treat the multi-view consistency in image editing through simple changes in DDPM sampling avoiding expensive 3D optimization.
2. Extended analysis: interesting ablation studies on backbone variations (Figure 8), coupled sampling on different prompts (Figure 9), and guidance strength analysis (Figure 10) which give insights into the method's behavior.
3. Efficient method: when comparing with optimization-based methods such as NeRF/3DGS, the model is significantly faster and demonstrated to work with sparse views.
4. Thorough validation: the paper presents results on three editing tasks (spatial editing, stylization, relighting) as well as a  user studies with 25 participants. The method is also shown to be generalisable across different model architectures (MVDream, MV-Adapter, Stable Virtual Camera) and diffusion backbones (SD1.5, SD2.1, SDXL, Flux) suggesting potential for broader applications.

**Weaknesses:**

1. Theoritical basis. The paper is lacking in rigorous theoritical justification for the presented method. Line 192, a clear justification of the relationship between the constant coefficient lambda and the consistency evaluation is missing.

2. Limited analysis of failure cases. Failure cases and insights into when the method would fail is not discussed.

3. Computational overhead. There are several mentioning of the reduced computational overhead by the presented method but no direct comparisons with state-of-the-art methods is given.

4. Framework characteristics. The scalability of the approach is not mentioned.

5. Limitations. The authors state that the "edited outputs are not perfectly 3D consistent compared to test-time optimization-based method" (line 648). However, there is no quantification of the consistency here.

**Questions:**

1. What makes this coupling mechanism robust to noise?  Flickering artifcats are shown in Figure 6 for comparison methods but it is not clear why the presented method would be exempt from artifacts.

2. The constant coefficient lambda for the guidance strength seems to be arbitrarily defined. What is the reasoning for the selection of this parameter?

3. For the coupling mechanism, Euclidean distance is selected. Have you explored different distance-based methods ?

4. Have you experimented with more than 21 frames? How scalable is your method?

5. What is the reduction in consistency compared to optimization-based methods? Could you please quantify the consistency loss? The artifacts in Figure 6 for qualitative visualisation do not give a clear view of what is happening.

---

> ### Author Response · Authors · 2025-11-14
> **Thanks for the review**
>
> Thanks for the thoughtful review.
>
> - The choice of 21 frames is simply based on the fact that Stable-Virtual-Camera was trained to generate 21 frames simultaneously. This is independent of our method, as it can be run on any number of views that the base models accept. In the supplementary materials, we include varying multi-view diffusion models that operate on 4 and 6 frames as well.
> - The coupling strength coefficient $\lambda$ is a hyper-parameter. When setting it to 0, we obtain standard DDPM sampling, and as we increase it, the output samples start to follow the guidance objective. However, as in prior guidance techniques, setting the guidance too large can cause the samples to eventually collapse and produce artifacts. Similar to prior work, the ideal value is the highest value possible without collapsing.
>
> Thanks for the constructive review and we will incorporate the feedback in the revision.

---

### Official Review · Reviewer_4M4i · 2025-10-30

**Soundness:** 3
**Presentation:** 3
**Contribution:** 2
**Rating:** 6
**Confidence:** 5

**Summary:**

This paper introduces a trainig-free method for multi-view consistent image editing using pre-trained 2D image editing models and pre-trained multi-view diffusion models. The proposed coupled diffusion sampling enables each view to be edited consistently across spatial editing, stylization, and relighting tasks. Experimental results show clear improvements over baselines in both quantitative metrics and human evaluation.

**Strengths:**

1. The problem addressed is practical and the proposed method is simple yet effective without additional training.
2. The performance is promising and surpasses baselines approaches.
3. The paper is clearly written and easy to follow.

**Weaknesses:**

1. The theoretical analysis is rather shallow and mostly intuitive.
2. The paper is missing two recent related works on training-free 2D image editing:
 1. FlowEdit: Inversion-Free Text-Based Editing Using Pre-Trained Flow Models, ICCV 2025
 2. Highly Consistent and Precise Training-free Visual Editing, SIGGRAPH Asia 2025
3. Although limitations are discussed in the appendix, it would be helpful to visualize common failure cases.
4. The relighting performance appears weaker compared to spatial editing and stylization. The background consistency in Figures 1 and 7 seems worse. Clarification on this would strengthen the paper.
5. In the caption of Figure 8, “multli-view” should be corrected to “multi-view.”

**Questions:**

1. Could the authors provide visualizations of typical failure cases?
2. For relighting, what might explain the performance drop compared to the other two tasks?

---

> ### Author Response · Authors · 2025-11-14
> **Thanks for the review**
>
> Thanks for the positive review.
>
> - We will include visualizations of failure cases in the revision. The failure cases fall into two categories: 1) limitations of the base model (for example, if a 3D scene is out of domain for the multi-view generative model and the sampling fails, adding guidance cannot help recover from that). 2) similar to prior guidance methods, setting the guidance coefficient too large can cause the samples to collapse, which can produce samples outside of the multi-view distribution.
> - The background consistency suffers due to a combination of: stable virtual camera prior on backgrounds can be limited, and IC-Light rely on the synthesized background to dictate the object lighting. However, we believe this can be resolved with spatially aware coupling.
>
> Thanks for the constructive review, and we will incorporate the feedback in the revision.

---

### Note · Authors · 2025-11-14

**Comment:**

Thanks for all the reviewers for the constructive comments, and we will incorporate the feedback in our revision.

**Withdrawal Confirmation:**

I have read and agree with the venue's withdrawal policy on behalf of myself and my co-authors.